# Factors affecting bus accident severity in Thailand: A multinomial logit model

**Wiriya Mahikul**[1], **Ongvisit Aiyasuwan**[2], **Pashanun Thanartthanaboon**[2], **Wares Chancharoen**[1], **Paniti Achararit**[1], **Thakdanai Sirisombat**[3], **Phathai Singkham**[2]*

1 Princess Srisavangavadhana College of Medicine, Chulabhorn Royal Academy, Bangkok, Thailand,
2 Division of Innovation and Research, Department of Disease Control, Ministry of Public Health, Nonthaburi, Thailand, 3 Institute of Field Robotics, King Mongkut's University of Technology Thonburi, Bangkok, Thailand

* phathais@gmail.com

**Data Availability Statement:** All relevant data are within the paper and its Supporting Information files.

**Funding:** This work was partly funded by Department of Disease Control, Ministry of Public

## Abstract

Bus accidents are a serious issue, with high rates of injury and fatality in Thailand. However, no studies have been conducted on the factors affecting bus accident severity in Thailand. A cross-sectional study was conducted by the Department of Highways, Thailand over the 2010–2019 period. A multinomial logit model was used to evaluate the factors associated with bus accident severity. This model divided accidents into three categories: non-injury, injury, and fatality. The risk factors consisted of three major categories: the bus driver, characteristics of the crash, and environmental characteristics. The results showed that characteristics of the bus driver, the crash, and the environment where the crash occurred all increased the probability of bus accidents causing injury. These three main factors included driving on sloped roads (relative risk ratio [RRR] 3.03, 95% confidence level [CI] 1.73 to 5.30), drowsy driving (RRR 2.60, 95% CI 1.71 to 3.96), and driving in the wrong direction (RRR 2.37, 95% CI 1.77 to 3.19). Moreover, the factors that increased the probability of the accidents causing fatality were drowsy driving (RRR 3.40, 95% CI 2.07 to 5.57) and drivers not obeying or following traffic rules (RRR 3.02, 95% CI 1.95 to 4.67), especially in the northern part of Thailand (RRR 3.01, 95% CI 1.98 to 4.62). The results can provide a valuable resource to help road authorities in development targeting road safety programs at sloped roads in the northern part of Thailand. Stakeholders should increase road safety efforts and implement campaigns, such as raising public awareness of the risks of not obeying or following traffic rules and drowsy driving which could possibly reduce the risk of both injury and fatality.

## Introduction

In developed countries such as the United States and Canada, buses are considered to be safe [1, 2]. However, the situation in developing countries may be different [3]; in Thailand, approximately 300 bus accidents are reported by the Department of Land Transport each year, with about 2,000 injuries and 200 fatalities per year [4]. According to World Health

Health, Thailand (FFB640004). The funders had no role in study design, data collection and analysis, decision to publish, or preparation of the manuscript.

**Competing interests:** The authors have declared that no competing interests exist.

Organization (WHO) estimates, the road traffic mortality rate including motorcyclists, car drivers, and passengers in Thailand remains at 32.2 per 100,000 population [5], which is ranked 1[st] in Asia in terms of crash fatality rates. While the rate of death among the drivers and passengers of buses is low (2%), bus accidents are a serious public health problem in terms of the property loss and personal damage caused by the accidents [6].

Studies on bus accidents are limited, and the factors associated with the severity of bus accident need to be addressed [7]. Several studies have been conducted on the safety of school buses [8, 9], and a few studies have analyzed the main risk factors associated with the occurrence of bus accident severity [1, 6, 10]. Risk factors comprise bus driver factors (such as not obeying or following traffic rules, abrupt driving, drunk driving, and falling asleep while driving), vehicle condition (such as the age of the bus and engine defects), the nature of the crash (such as the time of day and regionality), and the road factors (environmental factors such as road curves, median openings, road junctions, and highways without frontage roads). In terms of bus driver factors, several studies have examined their association with the severity of the outcome. For instance, in Iran, a study showed that bus accident rates were associated with drivers' sleep disorders [11], while in the US, the socio-economic of bus driver and speeding were found to be associated with bus fatalities [12]. In Sweden, several studies investigated the correlation between bus accident rates and driver acceleration behavior [13–15]; the studies showed that the correlations were strong enough to confirm that acceleration behavior can be a predictive variable for bus accidents. In less economically developed countries such as Ghana, a study showed that the factors associated with fatalities on the highway included speeding, wrongful overtaking, careless driving, and a lack of experience by drivers [16]. In Asia, several studies have been conducted. In Sri Lanka, the drivers' personal characteristics, such as high working hours and low salaries, were associated with private bus accidents [17], while in Thailand, the factors associated with the sleep quality among Thai intercity bus drivers were evaluated [18], but the study did not investigate its association with accident severity. In terms of crash characteristics, the time of day [19, 20], month [21], and region [22] where the crash occurred were associated with the severity of the outcome. These results also indicated that accidents occurring during peak evening hours were associated with a higher rate of fatality [23]. However, some studies have shown that crashes occurring during the morning rush hour were also associated with serious injury [23]. In the southern region of the U.S., the likelihood of light injuries and fatalities from bus accidents was lower than all other regions [1]. Additionally, in Iran, long holidays were significantly associated with the rate of accidents [24], which is similar to Thailand; there are usually higher incidences of road traffic injury over Thai long holidays such as over the New Year, in January, and during Songkran, in April [21].

As for road environmental factors, different kinds of road are known to affect the rate of bus accidents. For instance, in Canada, a study found that the factors associated with bus accidents were average daily traffic, public transportation and pedestrian traffic volumes, and transit characters [25]. In Denmark, the severity of bus accidents was associated with the weight of the vehicle, crossing intersections during a yellow or red light, open areas, and slippery road surfaces [19]. In Vietnam, a study revealed that the severity of the outcome increases due to rain, sparse traffic, the accident occurring in the evening or at night, or the accident occurring in a rural area, on roads with at least three lanes, or on curved roads or two-way roads without a physical barrier [26]. In term of vehicle features, a study in the US showed that accidents involving older buses caused more injuries and fatalities than those involving newer buses [6], and that road traffic accidents were associated with engine defects [27], while bus size was associated with the rate of bus accidents [10, 26]. The injury and fatality rate among school buses were lower than for other vehicle accidents [28].

Most studies divided the severity of accident into three categories: property damage only, injury, and fatality [23, 26, 29, 30]. However, there is little specialized research into injury and fatal accident severity, especially for bus accidents in Thailand. Therefore, this study aims to investigate the risk factors associated with bus accident severity in Thailand.

## Methodology

### Study design and data source

This study is cross-sectional study. Bus accident data were retrieved from the Department of Highways (DOH), Thailand over the 2010–2019 period [31]. Under the DOH composed of 18 offices of highways across the country, followed by "highway districts" composed of 106 districts [32]. Highways are main roads allocated for the transportation of people or goods that operating on urban and rural roads. It connects different regions, provinces, and districts [32]. The data were cleaned, and the missing information was removed from records; 2,911 events of bus accidents were used in this analysis. Some data were received immediately at scene; however, other data were gained afterward. Therefore, the data might not be representative of the real situation, with some of the accident causes missing. Based on a previous study [1], bus driver factors and environmental characteristics were classified as risk factors. A bus driver's gender and age were not analyzed in this study because of the lack of availability of this information in the dataset.

### Ethics approval

This study does not involve human participants. The study was approved by the Ethics Review Committee for Research in Human Subjects (Project Number Approval 64010), Thailand Ministry of Public Health, Department of Disease Control.

### Bus driver factors

Drunk driving, drowsy driving, sleep, driving at excessive speeds, abrupt driving, and not obeying traffic rules by the bus driver were considered as potential risk factors affecting bus accident severity. These certain variables were classified into two groups as "Yes", for cases where the bus driver displayed proper driving behavior, or "No", for cases where the bus driver lacked sleep or disobeyed traffic rules through actions such as driving drunk or driving at excessive speeds. Based on the previous research literature, a bus driver's behavior was divided two categories [1, 19, 20]. In the U.S., wrong-way accidents were associated with the severity of outcomes [20]; as a result, this study also classified the driving direction into three categories: unknown, right-way, and wrong-way.

### Crash characteristics

Previous studies in Thailand over long holidays such as the New Year's (in January) and Songkran holidays (in April) showed that there are usually higher road traffic accident and injury [21]. Therefore, our study divides the months into twelve categories related with Thai holidays. Time of day was also considered in this study. In previous study, the time of the bus accident was classified into four categories depending on traffic volume [19, 20]: early morning (6.00–9.00 a.m.), morning and afternoon (9.00 a.m.-3.00 p.m.), evening (3.00–6.00 p.m.), and night (6.00 p.m.-6.00 a.m.). However, our study grouped the time of the bus accident into three categories: morning (6.00–11.59 a.m.), afternoon and evening (12.00 p.m.-5.59 p.m.), and night (6.00 p.m.-5.59 a.m.). In the U.S., the probability of injury was reduced in the South region compared to other regions [1]. In Thailand, the estimated transport accident fatality

rates were highest in the central region and lower northern region [22]. Therefore, the accidents in this study were divided into six regions: Central, Northern, Northeast, Eastern, Western, and Southern as shown in Fig 1.

### Environmental characteristics

Based on previous studies, accidents on highways were associated with greater hospitalization [33]. A study in Malaysia showed that increased accident severity was associated with poor horizontal alignment [34]. The road's horizontal alignment in this study was categorized into three groups: straight, curved, and sharply curved. A previous study in China showed that vertical alignment was also associated with single-vehicle accidents [35]; therefore, in this study, road vertical alignment was divided into four categories: flat, upwardly curved, downwardly curved, and sloped. The type of intersection was associated with crash severity [23]. In this study, the type of intersection was divided into five groups, including no intersection, four-way, T-shaped, Y-shaped, and others. Previous studies showed that access points or median openings are critical locations that influence safety performance [36]; for example, in Croatia, more people were injured than killed at urban junctions [37].

### Statistical approach

The factors associated with the severity of bus accident in Thailand were evaluated by a multinomial logit model (MNL). Bus accident severity was divided into three categories: non-injury (property damage only), injury (hospitalization), and fatality. A 'non-injury' means an accident which leads to property damage only without any injury cases. An 'injury' means an accident which leads to the injury and needs hospitalization of a victim on the bus at least one case. A 'fatal accident' means an accident which leads to the death of a victim at least one death. Previous studies [1, 38] used the ordered logit or ordinal probit model to analyze the severity outcome, which is characterized by natural ordering (from a low level to a high level of severity). However, some limitations were found from these models [39]. The increased probability of the severest class was related to a decreased probability of the least severe class in an ordered model [40]. Therefore, it is not always suitable to analyze an ordered model for an ordered outcome [41]. Hence, the multinomial logit model has been recommended to be used for evaluation of the severity outcomes [16, 42]. MNL extends the binary logit model for situations where the independent variable has more than two categories [43]. The nominal category of the response variable was transformed into a numerical scale [44]. Let $Y = (Y_2, \ldots, Y_I)$ be a vector of dummy variables given that category 1 is the reference value. $Y$ is a random variable with a multinomial distribution, where $Y \sim \text{MN}(p, 1)$, and probability mass function in (1):

$$f(Y) = \frac{1}{\prod_{i=1}^{I} Y!} \prod_{i=1}^{I} p_i^{Y} \tag{1}$$

Where $p = (p_1, \ldots, p_I)$ is a vector of probabilities of bus driver experiencing the accident.

For a set of $n$ explanatory variables, denoted by $x = (x_1, \ldots, x_n)$ in (2):

$$p_i(x) = P(Y = i|x), i = 1, 2, \ldots \ldots, I, \tag{2}$$

which is a multinomial probability such that $\sum_{i=1}^{I} p_i(x) = 1$.

The multinomial logit model that is used in the study is presented below [23] in (3):

$$p_{ni} = \frac{e^{\beta_i X_{ni}}}{\sum_{i=1}^{I} e^{\beta_i X_{ni}}}, i = 1, 2, \ldots \ldots, I, \tag{3}$$

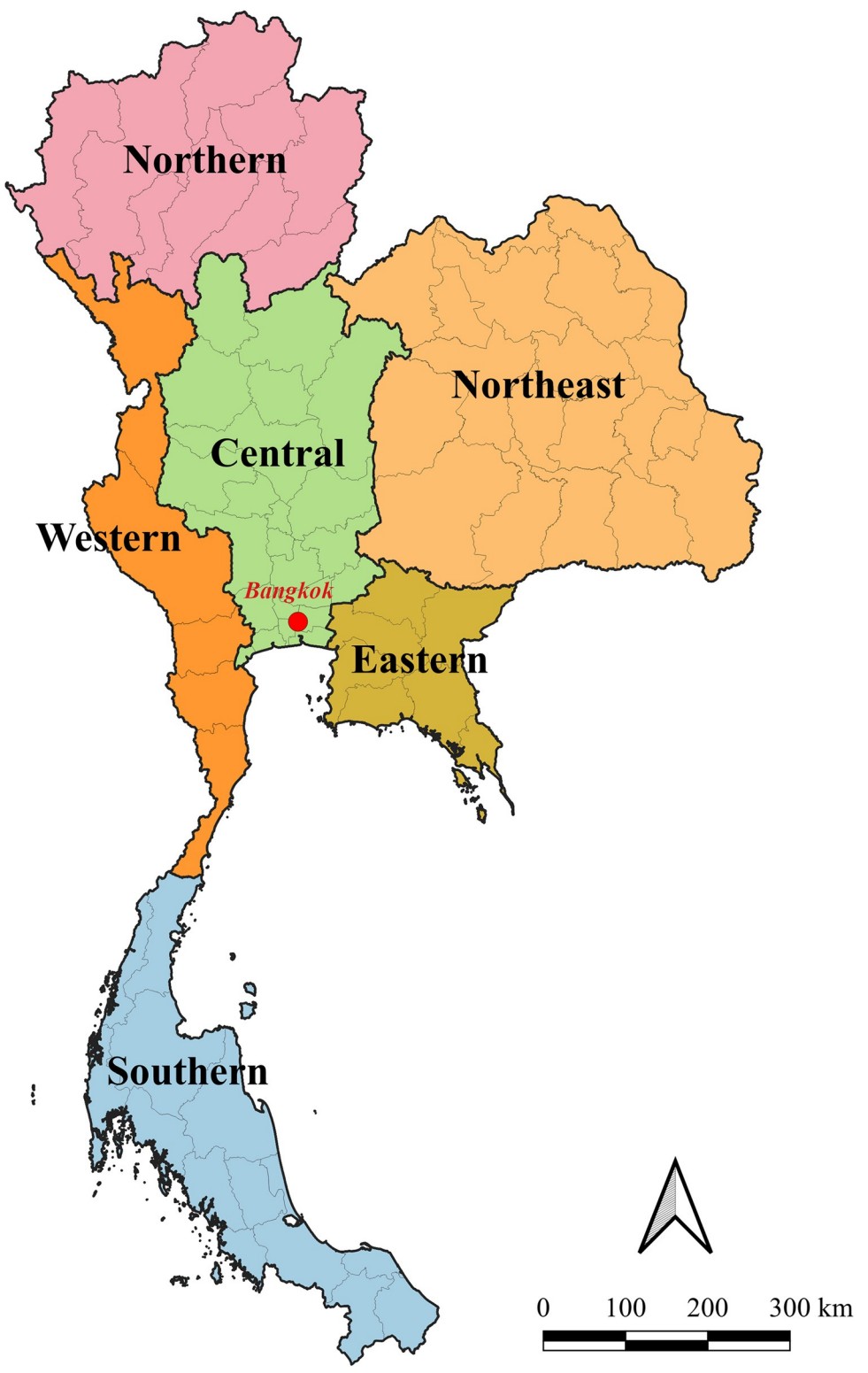

**Fig 1. Map of Thailand illustrating 6 regions.**

**Table 1. Independence of Irrelevant Alternatives (IIAs) property test.**

| Omitted severity level | $x^2$-Value | p-value | Null hypothesis | IIA property |
|---|---|---|---|---|
| Non-injury | 26.960 | 0.213 | Fail to reject | Holds |
| Injury | 25.800 | 0.260 | Fail to reject | Holds |
| Fatality | 27.053 | 0.209 | Fail to reject | Holds |

where $p_{ni}$ is the probability of the bus driver $n$ experiencing the severe injury of i, $\beta_i$ is a coefficient of the accident severity i, and $X_{ni}$ is an explanatory variable.

The maximum likelihood approach was used to estimate $\beta_i$. The non-injury from bus accident category, which had the highest reporting rate, was used as the reference category. According to the nonlinear assumptions of the multinomial logit model, the estimated coefficients is not suitable for indicating the direct effect on the severity outcomes; therefore, the relative risk ratio (RRR) was used to evaluate the risk factors which is computed relative to the base category. The relative probability of severity outcome (i = 2) to the non-injury category (i = 1) is

$$RRR = \frac{\Pr(i = 2)}{\Pr(i = 1)} = e^{\beta_i}. \tag{4}$$

The RRR represents the risk or protective factor for bus accident severity. When an RRR greater than 1, it shows an increase in risk, while an RRR lower than 1, it shows a decrease in the risk of a specific injury in terms of severity level compared with the base category [45]. In this study, the MNL and the associated RRR at the 95% confidence level were estimated using Stata (version 14.0). Marginal effects were computed and used to assess the effect of each variable on the bus accident severity outcome probabilities.

The independence of irrelevant alternatives (IIAs) for the response group is the assumption of the MNL [46]. The probability ratio was assumed to be the same for both levels of bus accident severity. The Hausman-McFadden test was used to evaluate the IIA [23, 47] as shown in Table 1. There is no reason to reject the null hypothesis or independence of irrelevant alternatives given the p-value is greater than 0.05. Therefore, the MNL model can be used in this study based on the certainty that the groups are independent of each other.

## Results

The count and percentage of the factors which might associated with the level of bus severity accident are represented in Table 2. More than half of bus driver (68.5%) was reported having excessive speed. The percentage of bus driver who not obeying traffic rules was similar for each category approximately 30.0%. Only 5–10% of bus driver was reported having abrupt driving and fallen asleep. However, drunk driving was reported with the high proportion of accident causing fatality (60.0%). In addition, the proportions of wrong way driving direction were 27.0%. The largest proportion of accident was reported in April (13.8%) and morning rush hour (50.4%). The central of Thailand was reported the high proportion of bus accident (44.5%). However, in the norther part of Thailand was reported the large percentage of bus accidents causing injury (30.6%). Two-third of accident was found at highways without a frontage road (74.6%). While one-third of accident causing fatality was found at curve road and Y-shaped road approximately 30.0%. About half of the bus accident causing injury was reported at median opening (43.5%) and road junctions (59.1%).

The results of the MNL model's estimation are shown in Table 3. In this table, the only statistically significant variable is presented. The estimated coefficient using maximum likelihood

**Table 2. Descriptive statistics (%) of risk factors from 2010 to 2019 in Thailand.**

| Variables | Non-injury | Injury | Fatality | Total |
|---|---|---|---|---|
| Bus driver factors | | | | |
| Excessive speed | | | | |
| No[a] | 32.1 | 40.6 | 27.3 | 31.5 |
| Yes | 47.0 | 38.1 | 14.9 | 68.5 |
| Not obeying traffic rules | | | | |
| No[a] | 42.9 | 39.2 | 17.9 | 94.8 |
| Yes | 30.9 | 32.9 | 36.2 | 5.2 |
| Abrupt driving | | | | |
| No[a] | 43.4 | 38.9 | 17.7 | 88.3 |
| Yes | 33.5 | 39.1 | 27.4 | 11.7 |
| Drunk driving | | | | |
| No[a] | 42.3 | 40.0 | 18.7 | 99.7 |
| Yes | 30.0 | 10.0 | 60.0 | 0.3 |
| Fallen asleep | | | | |
| No[a] | 43.6 | 38.0 | 18.4 | 94.4 |
| Yes | 20.9 | 53.4 | 25.7 | 5.6 |
| Driving direction | | | | |
| Unknown [a] | 55.9 | 27.5 | 16.6 | 13.9 |
| Right-way | 41.7 | 38.8 | 19.5 | 59.1 |
| Wrong-way | 36.5 | 44.9 | 18.6 | 27.0 |
| Crash characteristics | | | | |
| Month | | | | |
| January[a] | 40.5 | 40.9 | 18.6 | 11.3 |
| February | 43.8 | 36.4 | 19.8 | 7.4 |
| March | 40.4 | 36.5 | 23.1 | 8.9 |
| April | 40.9 | 40.5 | 18.6 | 13.8 |
| May | 48.1 | 35.7 | 16.2 | 7.4 |
| June | 47.8 | 38.5 | 13.7 | 6.3 |
| July | 41.1 | 40.1 | 18.8 | 6.9 |
| August | 43.7 | 38.0 | 18.3 | 7.7 |
| September | 52.7 | 35.5 | 11.8 | 5.8 |
| October | 39.1 | 38.1 | 22.8 | 6.8 |
| November | 43.6 | 40.4 | 16.0 | 7.5 |
| December | 33.9 | 42.4 | 23.7 | 10.1 |
| Time of day | | | | |
| Morning[a] | 40.2 | 41.0 | 18.8 | 50.4 |
| Afternoon/evening | 47.2 | 37.7 | 15.1 | 26.1 |
| Night | 41.5 | 35.6 | 22.9 | 23.5 |
| Region | | | | |
| Central[a] | 57.2 | 31.6 | 11.2 | 44.5 |
| Northern | 25.4 | 44.0 | 30.6 | 8.0 |
| Northeast | 30.3 | 46.5 | 23.3 | 22.5 |
| Eastern | 40.1 | 39.2 | 20.7 | 7.8 |
| Western | 26.1 | 46.6 | 27.3 | 5.5 |
| Southern | 29.1 | 45.0 | 25.9 | 11.7 |
| Environmental characteristics | | | | |
| Highways without a frontage road | | | | |

(*Continued*)

**Table 2.** (Continued)

| Variables | Non-injury | Injury | Fatality | Total |
|---|---|---|---|---|
| No[a] | 60.2 | 30.9 | 8.9 | 25.4 |
| Yes | 36.2 | 41.6 | 22.2 | 74.6 |
| Road horizontal alignment | | | | |
| Straight[a] | 44.9 | 38.5 | 16.6 | 85.5 |
| Curved | 27.4 | 41.0 | 31.6 | 13.4 |
| Sharply curved | 18.2 | 45.4 | 36.4 | 1.1 |
| Road vertical alignment | | | | |
| Flat[a] | 44.1 | 38.3 | 17.6 | 92.6 |
| Upwardly curved | 25.0 | 33.9 | 41.1 | 1.9 |
| Downwardly curved | 27.8 | 33.3 | 38.9 | 1.2 |
| Sloped | 15.2 | 56.0 | 28.8 | 4.3 |
| Intersection type | | | | |
| No Intersection[a] | 43.3 | 38.9 | 17.8 | 88.5 |
| Four-way | 33.8 | 40.0 | 27.2 | 4.7 |
| T-shaped | 34.9 | 38.8 | 26.3 | 4.4 |
| Y-shaped | 26.9 | 38.5 | 34.6 | 0.9 |
| Others | 37.8 | 37.8 | 24.4 | 1.5 |
| Median opening | | | | |
| No[a] | 43.1 | 38.5 | 18.4 | 92.6 |
| Yes | 31.9 | 43.5 | 24.5 | 7.4 |
| Road junctions | | | | |
| No[a] | 42.7 | 38.6 | 18.7 | 98.1 |
| Yes | 22.2 | 51.9 | 25.9 | 1.9 |

This table displays percentage. The total column contains percentages calculated across rows within one variable. The injury severity columns contain percentages calculated across columns within one row.

[a] Indicates referent category.

approach explains the differences compared to the non-injury outcome. The results show that the factors increasing the probability of injury were the driver falling asleep; driving in the right- or wrong-way direction; driving in the northern, northeast, western, or southern regions; driving on highways without frontage roads; or driving on sloped roads or at road junctions. In addition, the factors increasing the chance of fatality were similar to those causing injury, except for the driver not obeying traffic rules, abrupt driving, drunk driving, driving in the eastern region, or driving at median openings. Also, the results showed that only driving during the afternoon hours can reduce the probability of fatal injuries.

## Bus driver factors

The results show that bus drivers who do not obey traffic rules are 3.02 times (95% CI: 1.95–4.67) more likely to cause a fatality than a non-injury during a bus accident. The results revealed that bus drivers who are drunk while driving are 4.79 times (95% CI: 1.05–21.94) more likely to cause a fatality than a non-injury in bus accidents. There was no association between driving at excessive speeds and the severity of outcome in this study. The results of the model also showed that bus drivers who attempted to overtake another vehicle or displayed abrupt maneuvers were 2.48 times (95% CI: 1.79–3.43) more likely to cause a fatality compared to a non-injury. Accidents occurring due to the driver falling asleep or experiencing fatigue

**Table 3. The results of the MNL model for bus accidents in Thailand.**

| Variable | Estimated coefficient[a] | P-value | RRR[b] |
|---|---|---|---|
| Intercept [IJ] | -1.465 (0.157) | <0.001 | - |
| Intercept [FT] | -2.687 (0.214) | <0.001 | - |
| Bus driver factors: Not obeying traffic rules [FT] | 1.103 (0.223) | <0.001 | 3.02 (1.95–4.67) |
| Bus driver factors: Abrupt driving [FT] | 0.909 (0.166) | <0.001 | 2.48 (1.79–3.43) |
| Bus driver factors: Drunk driving [FT] | 1.568 (0.775) | 0.043 | 4.79(1.05–21.94) |
| Bus driver factors: Fallen asleep [IJ] | 0.957 (0.315) | <0.001 | 2.60 (1.71–3.96) |
| Bus driver factors: Fallen asleep [FT] | 1.225 (0.251) | <0.001 | 3.40 (2.07–5.57) |
| Bus driver factors: Right-way driving direction [IJ] | 0.505 (0.134) | <0.001 | 1.65 (1.27–2.15) |
| Bus driver factors: Wrong-way driving direction [IJ] | 0.867 (0.150) | <0.001 | 2.37 (1.77–3.19) |
| Bus driver factors: Wrong-way driving direction [FT] | 0.339 (0.188) | 0.034 | 1.49 (1.03–2.16) |
| Crash characteristics: Afternoon/evening [IJ] | -0.208 (0.104) | 0.045 | 0.95 (0.76–1.19) |
| Crash characteristics: Afternoon/evening [FT] | -0.374 (0.138) | 0.007 | 0.69 (0.52–0.90) |
| Crash characteristics: Northern region [IJ] | 0.784 (0.188) | <0.001 | 2.19 (1.51–3.16) |
| Crash characteristics: Northern region [FT] | 1.107 (0.216) | <0.001 | 3.01 (1.98–4.62) |
| Crash characteristics: Northeast region [IJ] | 0.757 (0.119) | <0.001 | 2.13 (1.68–2.69) |
| Crash characteristics: Northeast region [FT] | 0.934 (0.153) | <0.001 | 2.54 (1.88–3.43) |
| Crash characteristics: Eastern region [FT] | 0.617 (0.215) | 0.004 | 1.85 (1.21–2.83) |
| Crash characteristics: Western region [IJ] | 0.704 (0.214) | <0.001 | 2.02 (1.33–3.07) |
| Crash characteristics: Western region [FT] | 1.030 (0.251) | <0.001 | 2.80 (1.71–4.58) |
| Crash characteristics: Southern region [IJ] | 0.727 (0.156) | <0.001 | 2.07 (1.52–2.81) |
| Crash characteristics: Southern region [FT] | 0.973 (0.190) | <0.001 | 2.65 (1.82–3.84) |
| Environmental characteristics: Highways without frontage roads [IJ] | 0.546 (0.112) | <0.001 | 1.72 (1.38–2.15) |
| Environmental characteristics: Highways without frontage roads [FT] | 0.897 (0.164) | <0.001 | 2.45 (1.77–3.38) |
| Environmental characteristics: Curve roads [FT] | 0.595 (0.177) | 0.001 | 1.81 (1.28–2.56) |
| Environmental characteristics: Sloped roads [IJ] | 1.109 (0.285) | <0.001 | 3.03 (1.73–5.30) |
| Environmental characteristics: Sloped roads [FT] | 1.033 (0.322) | 0.001 | 2.81 (1.49–5.27) |
| Environmental characteristics: Median openings [FT] | 0.414 (0.204) | 0.042 | 1.51 (1.01–2.26) |
| Environmental characteristics: Road junctions [IJ] | 0.781 (0.362) | 0.031 | 2.18 (1.07–4.44) |
| Number of observations: 2,911 | | | |
| Log-likelihood: -2792.9203 | | | |
| Chi-square: 501.80 | | | |
| McFadden R-squared: 0.0824 | | | |
| P-value: 0.0000 | | | |

[IJ], injury; [FT], fatality. The reference category is the non-injury.

[a] Standard errors are in parentheses.

[b] Lower and upper limits at the 95% confidence interval (CI) are in parentheses.

were 2.60 times (95% CI: 1.71–3.96) and 3.40 times (95% CI: 2.07–5.57) more likely to result in injury or fatality compared to non-injury, respectively. Accidents occurring from the driver driving in the wrong direction were 2.37 times (95% CI: 1.77–3.19) and 1.49 times (95% CI: 1.03–2.16) more likely to result in injury or fatality compared to non-injury, respectively.

## Crash characteristics

Crashes occurring during peak morning traffic are 1.05 times (RRR = 1/0.95) and 1.45 times (RRR = 1/0.69) more likely to result in injury or fatality compared to non-injury, respectively. Crashes occurring in Northern Thailand are 2.19 times (95% CI: 1.51–3.16) and 3.01 times

**Table 4. The results of the marginal effects for bus accidents in Thailand (%).**

| Variables | Non-injury | Injury | Fatality |
|---|---|---|---|
| Bus driver factors | | | |
| Not obeying traffic rules | -11.4 | -8.4 | 19.7 |
| Abrupt driving | -11.5 | -1.5 | 13.1 |
| Drunk driving | -11.5 | -30.9 | 42.4 |
| Fallen asleep | -22.5 | 10.9 | 11.1 |
| Right-way driving direction | -10.0 | 10.7 | -0.7 |
| Wrong-way driving direction | -17.2 | 18.2 | -1.0 |
| Crash characteristics | | | |
| Afternoon/evening | 6.3 | -2.5 | -0.4 |
| Northern region | -19.4 | 8.4 | 11.1 |
| Northeast region | -18.5 | 10.5 | 8.1 |
| Eastern region | -9.9 | 2.8 | 7.1 |
| Western region | -17.8 | 7.2 | 10.6 |
| Southern region | -18.0 | 8.8 | 9.2 |
| Environmental characteristics | | | |
| Highways without frontage roads | -15.9 | 7.5 | 8.3 |
| Curve roads | -6.6 | -2.1 | 8.7 |
| Sloped roads | -22.6 | 17.2 | 5.5 |
| Median openings | -7.9 | 4.1 | 3.9 |
| Road junctions | -16.0 | 14.3 | 1.7 |

(95% CI: 1.98–4.62) more likely to cause in injury or fatality compared to non-injury, respectively.

## Environmental characteristics

Bus accidents that occur in highways without frontage roads are 1.72 times (95% CI: 1.38–2.15) and 2.45 times (95% CI: 1.77–3.38) more likely to result in injury or fatality compared to non-injury, respectively. The results show that bus accidents occurring on curved roads are 1.81 times (95% CI: 1.28–2.56) more likely to cause a fatality than non-injury. Bus accidents that occur on sloped roads are 3.03 times (95% CI: 1.73–5.30) and 2.81 times (95% CI: 1.49–5.27) more likely to result in injury or fatality compared to non-injury, respectively. The results show that bus accidents that occur at median openings are 1.51 times (95% CI: 1.01–2.26) more likely to cause a fatality than to a non-injury. Bus accidents that occur at road junctions are 2.18 times (95% CI: 1.07–4.44) more likely to result in a fatality than non-injury.

The results of the marginal effects for bus accidents in Thailand are given in Table 4. Drunk driving greatly increases the probability of fatal injury by 42.4% in bus accidents compared to accidents occurring in non-drunk driving. While, wrong-way driving direction increases the probability of injury by 18.2%. In terms of crash characteristics, the crashes occurring in Northern Thailand increases the probability of fatal injury by 11.1% in bus accidents compared to accidents occurring in Central region. Other crash characteristic variables show that the crashes occurring in Northeast Thailand increases the probability of injury by 10.5% compared to accidents occurring in Central region. Environmental characteristics were also found to be significant. Curve roads was found to significantly increase the probability of accident fatality by 8.7% compared to straight roads. While, sloped roads were found to significantly increase the probability of injury by 17.2% compared to flat roads.

## Discussion

### Bus driver factors

The study showed that bus drivers who do not obey traffic rules are more likely to cause a fatality than a non-injury during a bus accident. This result is consistent with previous studies [48], which demonstrated that bus drivers not obeying traffic rules tend to experience more severe injuries. The results revealed that bus drivers who are drunk while driving are more likely to cause a fatality than a non-injury in bus accidents. This result is consistent with a previous study [49], which reported that drunk driving among bus drivers is associated with more severe outcomes. In Thailand, the Ministry of Transport has implemented strict pre-departure screening to ban drunk bus drivers since 2011; however, accidents due to the bus drivers being intoxicated still occur [50]. There was no association between driving at excessive speeds and the severity of outcome in this study. In contrast, Prato CG *et al* [19] reported that high speed limits may increase bus accident severity. A recent study on road traffic injuries showed that drivers driving at excessive speeds were two to three times more likely to cause severe injuries [48, 51]. The results of the model also showed that bus drivers who attempted to overtake another vehicle or displayed abrupt maneuvers were more likely to cause a fatality compared to a non-injury, which is consistent with a previous study [48], which demonstrated that those who were abrupt driving were more than two times likely to be more severe injury. Accidents occurring due to the driver falling asleep or experiencing fatigue were more likely to result in injury or fatality compared to non-injury, respectively. Another study [52] showed that the rate of accident was associated with asleep and tired drivers [53, 54]; this indicates that driver sleepiness or tiredness is a problem for city bus drivers. Accidents occurring from the driver driving in the wrong direction were more likely to result in injury or fatality compared to non-injury, respectively. This result is consistent with a previous study [55], which reported that wrong-way driving crashes are associated with more severe outcomes. To prevent the bus driver risk factors, previous studies showed that awareness campaigns in reducing the accidents on highways should be implemented [56] and the driver should be encouraged to register the driver licensing [57].

### Crash characteristics

Crashes occurring during peak morning traffic are more likely to result in injury or fatality compared to non-injury, respectively. This is due to the fact that crashes occurring during the morning hours tend to be at higher speeds or are affected by the conditions of rush hour driving. This result is consistent with previous studies [19, 23, 58]. Crashes occurring in Northern Thailand are more likely to cause in injury or fatality compared to non-injury, respectively. This might be due to the fact that landslides occur in the upper northern region of Thailand, where the roads are mostly curved and the terrain is mountainous [59]. Moreover, Phayao was found to have the highest fatality rate (60.0%) of all provinces. In the U.S., regionality was associated with severity of outcome [1]. In Thailand, the estimated transport accident fatality rates were highest in the central region or lower northern region [22]. Previous studies showed that rescheduling of rush hour driving could reduce the incidence of crashes [60] and avoiding unfamiliar place could be reduced the risk of accident [61].

### Environmental characteristics

Bus accidents that occur in highways without frontage roads are more likely to result in injury or fatality compared to non-injury, respectively. This result is consistent with previous studies [33, 62], which reported that accidents on highways were associated with more severe

outcomes. The results show that bus accidents occurring on curved roads are more likely to cause a fatality than non-injury. These results are supported by previous studies [63], which reported that the most vulnerable sites for crash occurrence are on horizontally curved roads. Bus accidents that occur on sloped roads are more likely to result in injury or fatality compared to non-injury, respectively. Another study [1, 64] demonstrated that roads with curved and sloped alignments were associated with a greater probability of fatality. The results show that bus accidents that occur at median openings are more likely to cause a fatality than to a non-injury. Previous research [36] has also shown that median openings are associated with a higher risk of severe injury. Bus accidents that occur at road junctions are more likely to result in a fatality than non-injury. This result is consistent with previous studies [37, 65], which revealed that more people were injured than killed at urban junctions. To prevent bus accident severity, previous studies suggested that policy recommendations for improving road conditions should be proposed [26] and road authorities should concern the safety on the high risk roads [66–68].

## Limitations

This study only analyzed secondary data at the scene, which were limited to factors available from the Department of Highways database and may not fully explain the causal relationship. Additional factors such demographics, attitudes, and perceptions of bus drivers may help to create a more accurate and inclusive study. Previous studies showed that affecting parameters such as the red-light running behaviors [69], individual characteristics [70], and speed limit standards [71] on the injury severity outcomes may have heterogeneity and temporal instability across different time periods that could exist in the models. As such, random parameters model such as random parameters logit model with heterogeneity [72] and Bayesian approach [69, 70] could be used in further study. Moreover, there was no association between excessive speed and severity of outcome in this study; however, this might be caused by a limitation of the data on speed at the scene of the accident. Therefore, more studies utilizing cohort design, GPS, and detective devices to not only collect data from accidents but also measure the behavior of bus drivers and weather conditions could help evaluate the real risk factors associated with bus accidents.

## Conclusions

This study aimed to investigate the risk factors associated with bus accident severity in Thailand. To achieve this purpose, 2,911 bus accidents recorded by the Department of Highways in Thailand over the 2010–2019 period were analyzed using a multinomial logit model. The severity of bus accidents was divided into three categories: non-injury, injury, and fatality. Risk factors included bus driver factors (such as not obeying or following traffic rules, abrupt driving, driving in the wrong direction, drunk driving, and falling asleep), vehicle condition (such as age of the bus and engine defects), crash conditions (such as the time of day and regionality), and the road condition (environmental factors such as road curves, median openings, road junctions, and highways without frontage roads). The relative risk ratio (RRR) was used in this study to evaluate these risk factors; the results showed that the risk factors increasing the probability of injury included driving in the morning, driving in the wrong direction, driving on highways without a frontage road, driving on sloped roads, driving at road junctions, the driver falling asleep, or driving in the northern part of Thailand. Moreover, the factors that increased the likelihood of fatality were similar to those increasing the probability of injury, except for the curve of the road, driving at median openings or road junctions, pulling abrupt maneuvers while driving, drunk driving, and not obeying traffic rules by performing

actions such as turning off lights in the nighttime, illegally overtaking other vehicles, or driving through a red light.

The results from the study can provide a valuable resource to help road authorities in development targeting road safety improvements at road curves, median openings, road junctions and highways without frontage roads, as they may need to establish more traffic signs to warn bus drivers at these places. Roads and junctions should be wide and well-lit for the safety of the bus driver. Rapid notification, rescue accessibility, and availability of care should be prepared at highways without frontage roads, especially in the northern part of Thailand and during the morning rush hours. Stakeholders should increase road safety efforts and enact campaigns to promote road safety, such as raising public awareness of the risks of not obeying or following traffic rules, drowsy driving, driving in the wrong direction, drunk driving, and abrupt driving in order to reduce the possible risk of injury or fatality. Driving licenses should be strictly checked and continually renewed to ensure current traffic rules are understood by bus drivers. The results from this analysis obtained in different regions reduced the geographical bias in relation to bus accidents. To narrow the research, this study has exclusively focused on bus accident severity in a developing country, Thailand, which is important to inform the public awareness on bus accident.

## Supporting information

**S1 Data.**
(CSV)

## Acknowledgments

We thank the Department of Highways (DOH) and Ministry of Public Health, Thailand, for the data inputs. We gratefully acknowledge all experts especially Dr. Chamaiparn Santikarn.

## Author Contributions

**Conceptualization:** Wiriya Mahikul, Ongvisit Aiyasuwan, Pashanun Thanartthanaboon, Wares Chancharoen, Paniti Achararit, Thakdanai Sirisombat, Phathai Singkham.

**Data curation:** Ongvisit Aiyasuwan.

**Formal analysis:** Wiriya Mahikul.

**Funding acquisition:** Phathai Singkham.

**Investigation:** Wiriya Mahikul.

**Methodology:** Wiriya Mahikul.

**Project administration:** Phathai Singkham.

**Software:** Wiriya Mahikul.

**Validation:** Wiriya Mahikul.

**Writing – original draft:** Wiriya Mahikul.

**Writing – review & editing:** Wiriya Mahikul, Ongvisit Aiyasuwan, Pashanun Thanartthanaboon, Wares Chancharoen, Paniti Achararit, Thakdanai Sirisombat, Phathai Singkham.

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
