## [Decision Letter · Decision Letter 0]

22 Aug 2022

PONE-D-22-22254Factors affecting bus accident severity in Thailand: A multinomial logit modelPLOS ONE

Dear Dr. Singkham,

Thank you for submitting your manuscript to PLOS ONE. After careful consideration, we feel that it has merit but does not fully meet PLOS ONE’s publication criteria as it currently stands. Therefore, we invite you to submit a revised version of the manuscript that addresses the points raised during the review process.

Please submit your revised manuscript by September 20, 2022.  If you will need more time than this to complete your revisions, please reply to this message or contact the journal office at plosone@plos.org. Please include the following items when submitting your revised manuscript:A rebuttal letter that responds to each point raised by the academic editor and reviewer(s). You should upload this letter as a separate file labeled 'Response to Reviewers'.A marked-up copy of your manuscript that highlights changes made to the original version. You should upload this as a separate file labeled 'Revised Manuscript with Track Changes'.An unmarked version of your revised paper without tracked changes. You should upload this as a separate file labeled 'Manuscript'.

We look forward to receiving your revised manuscript.

Kind regards,

Yanyong Guo, Ph.D

Academic Editor

PLOS ONE

Journal Requirements:

"We thank the Department of Highways (DOH) and Ministry of Public Health, Thailand, for the data inputs."

"This work was partly supported by Department of Disease Control, Ministry of Public Health, Thailand (FFB640004). The funders had no role in study design, data collection and analysis, decision to publish, or preparation of the manuscript. "

Reviewers' comments:

Reviewer's Responses to Questions

**Comments to the Author**

1. Is the manuscript technically sound, and do the data support the conclusions?

Reviewer #1: Yes

Reviewer #2: Yes

2. Has the statistical analysis been performed appropriately and rigorously? 

Reviewer #1: Yes

Reviewer #2: No

3. Have the authors made all data underlying the findings in their manuscript fully available?

Reviewer #1: Yes

Reviewer #2: No

4. Is the manuscript presented in an intelligible fashion and written in standard English?

Reviewer #1: Yes

Reviewer #2: Yes

5. Review Comments to the Author

Reviewer #1: 1. the study is good performance with suitable methodology and valuable findings. The discussion is in-depth and compared with existing studies.

2. the discrete choice model family was commonly used in crash severity modeling, MNL is an variant of binary logit model when the independent variable is more than two levels. The transformation procedure could be provided in the literature.

3. Heterogeneous effect of variable could exist in the models. As such, random parameters model could be used in further study. At least, this could be discussed in the limitation. Related reference are as follows

Guo, Y., Li, Z., Wu, Y., & Xu, C. (2018). Exploring unobserved heterogeneity in bicyclists’ red-light running behaviors at different crossing facilities. Accident Analysis & Prevention, 115, 118-127.

Guo, Y., Li, Z., Wu, Y., & Xu, C. (2018). Evaluating factors affecting electric bike users’ registration of license plate in China using Bayesian approach. Transportation research part F: traffic psychology and behaviour, 59, 212-221.

Reviewer #2: This topic is interesting, and the whole manuscript is well organized, but it is very necessary to highlight its major contribution to the literature. I have the following issues that require particular attention in further revision:

1) It should be noted that affecting parameters on the injury severity outcomes may have temporal instability across different time periods. The authors use data from 2010 to 2019. This time period covers the great recession and other years where temporal instability of parameters is a known problem. During such a large time period that many influencing factors, such as the roadway network, population numbers, signal control conditions, speed limit standards, etc., have changed significantly, so it is not practical to disregard these changes for this study;

2) Some key indicators need to be clearly defined. For example, how do you distinguish between injury and fatality, and what are the specific criteria for morning, afternoon/evening and night? Why is it necessary to study by region (central, northern, northeast, etc.; a picture may be better to show the different region)?

3) Analysis of marginal effects is lacking and is recommended to be added;

4) What type of road does highway in Table 2 specifically refer to? This in turn suggests to us that the research object in this study, bus, refers only to that operating on urban roads? All these details need to be spelled out;

5) In the discussion section, it is suggested that the authors further suggest relevant preventive policy measures for bus accidents;

6) It is recommended that one form be retained for the number or percentage of accidents in Table 2; both p-values and asterisks are given in Table 3, and the information on both is repeated, so it is recommended that one form be retained.

6. PLOS authors have the option to publish the peer review history of their article (what does this mean?). If published, this will include your full peer review and any attached files.

Reviewer #1: No

Reviewer #2: No

---

## [Author Response · Author response to Decision Letter 0]

28 Sep 2022

Response to Reviewer 1 Comments

Comments and Suggestions for Authors

Reviewer #1: 

Point 1: 

1. the study is good performance with suitable methodology and valuable findings. The discussion is in-depth and compared with existing studies.

Response 1

Thank you for your kind comments.

Point 2:

2. the discrete choice model family was commonly used in crash severity modeling, MNL is an variant of binary logit model when the independent variable is more than two levels. The transformation procedure could be provided in the literature.

Response 2

We agreed with the reviewer. We have further reviewed and revised the transformation procedure of MNL in the literature as follow;

“MNL extends the binary logit model for situations where the independent variable has more than two categories [1]. The nominal category of the response variable was transformed into a numerical scale [2]. Let Y = (Y_2,…., Y_I) be a vector of dummy variables given that category 1 is the reference value. Y is a random variable with a multinomial distribution, where Y ∼ MN(p, 1), and probability mass function in (1):

f(Y)=1/(∏_(i=1)^I▒Y!) ∏_(i=1)^I▒p_i^Y (1)

Where p = (p_1,…, p_I) is a vector of probabilities of bus driver experiencing the accident.

 For a set of n explanatory variables, denoted by x = (x_1,…, x_n) in (2):

 p_i (x)=P(Y=i│x),i=1,2,……,I, (2)

 which is a multinomial probability such that ∑_(i=1)^I▒〖p_i (x)=1〗.

 The multinomial logit model that is used in the study is presented below [3] in (3):

p_ni=e^(β_i X_ni )/(∑_(i=1)^I▒e^(β_i X_ni ) ),i=1,2,…..,I, (3)

 where p_ni is the probability of the bus driver n experiencing the severe injury of i, β_i is a coefficient of the accident severity i, and X_ni is an explanatory variable.”

Page 6 Line 177

Point 3:

3. Heterogeneous effect of variable could exist in the models. As such, random parameters model could be used in further study. At least, this could be discussed in the limitation. Related reference are as follows

Guo, Y., Li, Z., Wu, Y., & Xu, C. (2018). Exploring unobserved heterogeneity in bicyclists’ red-light running behaviors at different crossing facilities. Accident Analysis & Prevention, 115, 118-127.

Guo, Y., Li, Z., Wu, Y., & Xu, C. (2018). Evaluating factors affecting electric bike users’ registration of license plate in China using Bayesian approach. Transportation research part F: traffic psychology and behaviour, 59, 212-221.

Response 3: 

We agreed with the reviewer and added the sentence in the limitation of this study to explain the heterogeneous effect of variable that could exist in the models as following;

“Previous studies showed that affecting parameters such as the red-light running behaviors [4], individual characteristics [5], and speed limit standards [6] on the injury severity outcomes may have heterogeneity and temporal instability across different time periods that could exist in the models. As such, random parameters model such as random parameters logit model with heterogeneity [7] and Bayesian approach [4,5] could be used in further study.”

Page 13 Line 423

Reference:

1. Withers, S.D. Data Analysis, Categorical. In International Encyclopedia of Human Geography (Second Edition), Kobayashi, A., Ed. Elsevier: Oxford, 2009; https://doi.org/10.1016/B978-0-08-102295-5.10364-6pp. 159-165.

2. Hashimoto, E.M.; Ortega, E.M.M.; Cordeiro, G.M.; Suzuki, A.K.; Kattan, M.W. The multinomial logistic regression model for predicting the discharge status after liver transplantation: estimation and diagnostics analysis. J Appl Stat 2020, 47, 2159-2177, doi:10.1080/02664763.2019.1706725.

3. Abrari Vajari, M.; Aghabayk, K.; Sadeghian, M.; Shiwakoti, N. A multinomial logit model of motorcycle crash severity at Australian intersections. Journal of Safety Research 2020, 73, 17-24, doi:https://doi.org/10.1016/j.jsr.2020.02.008.

4. Guo, Y.; Li, Z.; Wu, Y.; Xu, C. Exploring unobserved heterogeneity in bicyclists’ red-light running behaviors at different crossing facilities. Accident Analysis & Prevention 2018, 115, 118-127, doi:https://doi.org/10.1016/j.aap.2018.03.006.

5. Guo, Y.; Li, Z.; Wu, Y.; Xu, C. Evaluating factors affecting electric bike users’ registration of license plate in China using Bayesian approach. Transportation Research Part F: Traffic Psychology and Behaviour 2018, 59, 212-221, doi:https://doi.org/10.1016/j.trf.2018.09.008.

6. Alnawmasi, N.; Mannering, F. A statistical assessment of temporal instability in the factors determining motorcyclist injury severities. Analytic Methods in Accident Research 2019, 22, 100090, doi:https://doi.org/10.1016/j.amar.2019.100090.

7. Xie, P.; Qin, D.; Zhu, T. Impact of rule-violating behaviors on the risk of bus drivers being at-fault in crashes. Traffic Inj Prev 2022, 23, 364-368, doi:10.1080/15389588.2022.2079639.

Response to Reviewer 2 Comments

Comments and Suggestions for Authors

Reviewer #2: 

This topic is interesting, and the whole manuscript is well organized, but it is very necessary to highlight its major contribution to the literature. I have the following issues that require particular attention in further revision:

Point 1: 

1) It should be noted that affecting parameters on the injury severity outcomes may have temporal instability across different time periods. The authors use data from 2010 to 2019. This time period covers the great recession and other years where temporal instability of parameters is a known problem. During such a large time period that many influencing factors, such as the roadway network, population numbers, signal control conditions, speed limit standards, etc., have changed significantly, so it is not practical to disregard these changes for this study;

Response 1

Thank you for your kind comments. We agreed with the reviewer and added the sentence in the limitation of this study to explain the temporal instability across different time periods as following;

“Previous studies showed that affecting parameters such as the red-light running behaviors [1], individual characteristics [2], and speed limit standards [3] on the injury severity outcomes may have heterogeneity and temporal instability across different time periods that could exist in the models. As such, random parameters model such as random parameters logit model with heterogeneity [4] and Bayesian approach [1,2] could be used in further study.”

Page 13 Line 423

Point 2:

2. Some key indicators need to be clearly defined. For example, how do you distinguish between injury and fatality, and what are the specific criteria for morning, afternoon/evening and night? Why is it necessary to study by region (central, northern, northeast, etc.; a picture may be better to show the different region)?

Response 2

We agreed with the reviewer. The criteria to distinguish between injury and fatality is that the accident injury defined as people have been hospitalized at least one case, but fatality means people have died by the bus accident at least one death. The specific criteria for morning, afternoon/evening and night are that morning is defined as 6.00-11.59 a.m., afternoon and evening (12.00 p.m.-5.59 p.m.), and night (6.00 p.m.-5.59 a.m.). Those categories depend on traffic volume. Moreover, it is necessary to study by region (central, northern, northeast, etc.) because the bus accident varies across the region. We have further reviewed and revised the explanation of the severity outcome, time periods, and region.

Therefore, we added the sentence as follow:

“A ‘non-injury’ means an accident which leads to property damage only without any injury cases. An ‘injury’ means an accident which leads to the injury and needs hospitalization of a victim on the bus at least one case. A 'fatal accident' means an accident which leads to the death of a victim at least one death.”

Page 6 Line 167

We also added the Thai map to show the different regions as follow:

“Therefore, the accidents in this study were divided into six regions: Central, Northern, Northeast, Eastern, Western, and Southern as shown in Fig 1.”

Fig 1. Map of Thailand illustrating 6 regions.

Page 5 Line 148

Point 3:

3) Analysis of marginal effects is lacking and is recommended to be added;

Response 3: 

We agreed with the reviewer and analysed the marginal effects as following;

“Marginal effects were computed and used to assess the effect of each variable on the bus accident severity outcome probabilities.”

Page 7 Line 214

“The results of the marginal effects for bus accidents in Thailand are given in Table 4. Drunk driving greatly increases the probability of fatal injury by 42.4% in bus accidents compared to accidents occurring in non-drunk driving. While, wrong-way driving direction increases the probability of injury by 18.2%. In terms of crash characteristics, the crashes occurring in Northern Thailand increases the probability of fatal injury by 11.1% in bus accidents compared to accidents occurring in Central region. Other crash characteristic variables show that the crashes occurring in Northeast Thailand increases the probability of injury by 10.5% compared to accidents occurring in Central region. Environmental characteristics were also found to be significant. Curve roads was found to significantly increase the probability of accident fatality by 8.7% compared to straight roads. While, sloped roads were found to significantly increase the probability of injury by 17.2% compared to flat roads.

Table 4 

The results of the marginal effects for bus accidents in Thailand

Variables Non-injury Injury Fatality

Bus driver factors

Not obeying traffic rules -11.4% -8.4% 19.7%

Abrupt driving -11.5% -1.5% 13.1%

Drunk driving -11.5% -30.9% 42.4%

Fallen asleep -22.5% 10.9% 11.1%

Right-way driving direction -10.0% 10.7% -0.7%

Wrong-way driving direction -17.2% 18.2% -1.0%

Crash characteristics

Afternoon/evening 6.3% -2.5% -0.4%

Northern region -19.4% 8.4% 11.1%

Northeast region -18.5% 10.5% 8.1%

Eastern region -9.9% 2.8% 7.1%

Western region -17.8% 7.2% 10.6%

Southern region -18.0% 8.8% 9.2%

Environmental characteristics

Highways without frontage roads -15.9% 7.5% 8.3%

Curve roads -6.6% -2.1% 8.7%

Sloped roads -22.6% 17.2% 5.5%

Median openings -7.9% 4.1% 3.9%

Road junctions -16.0% 14.3% 1.7%

Page 11 Line 307

Point 4:

4) What type of road does highway in Table 2 specifically refer to? This in turn suggests to us that the research object in this study, bus, refers only to that operating on urban roads? All these details need to be spelled out;

Response 4: 

We agreed with the reviewer and explained the definition of the highway in Thailand which refers to that operating on urban and rural roads. Highways are main roads allocated for the transportation of people or goods that operating on urban and rural roads. It connects different regions, provinces, and districts.

Therefore, we added the sentences to define the highway as following;

“Highways are main roads allocated for the transportation of people or goods that operating on urban and rural roads. It connects different regions, provinces, and districts [5]. ”

Page 3 Line 110

Point 5:

5) In the discussion section, it is suggested that the authors further suggest relevant preventive policy measures for bus accidents;

Response 5: 

We agreed with the reviewer and added the relevant preventive policy measures for bus accidents in the discussion section as following;

“To prevent the bus driver risk factors, previous studies showed that awareness campaigns in reducing the accidents on highways should be implemented [6] and the driver should be encouraged to register the driver licensing [7].”

Page 13 Line 383

“Previous studies showed that rescheduling of rush hour driving could reduce the incidence of crashes [8] and avoiding unfamiliar place could be reduced the risk of accident [9].”

Page 13 Line 397

“To prevent bus accident severity, previous studies suggested that policy recommendations for improving road conditions should be proposed [10] and road authorities should concern the safety on the high risk roads [11-13].”

Page 13 Line 415

Point 6:

6) It is recommended that one form be retained for the number or percentage of accidents in Table 2; both p-values and asterisks are given in Table 3, and the information on both is repeated, so it is recommended that one form be retained.

Response 6: 

We agreed with the reviewer, however in table2, we presented both frequencies and percentage because it would be benefit for reader, in case, they need to recalculate the the frequencies to percentage in both row and column. While we deleted the asterisks in Table 3 to avoid the repeat of p-value. 

Page 10 Line 290

Reference:

1. Guo, Y.; Li, Z.; Wu, Y.; Xu, C. Exploring unobserved heterogeneity in bicyclists’ red-light running behaviors at different crossing facilities. Accident Analysis & Prevention 2018, 115, 118-127, doi:https://doi.org/10.1016/j.aap.2018.03.006.

2. Guo, Y.; Li, Z.; Wu, Y.; Xu, C. Evaluating factors affecting electric bike users’ registration of license plate in China using Bayesian approach. Transportation Research Part F: Traffic Psychology and Behaviour 2018, 59, 212-221, doi:https://doi.org/10.1016/j.trf.2018.09.008.

3. Alnawmasi, N.; Mannering, F. A statistical assessment of temporal instability in the factors determining motorcyclist injury severities. Analytic Methods in Accident Research 2019, 22, 100090, doi:https://doi.org/10.1016/j.amar.2019.100090.

4. Xie, P.; Qin, D.; Zhu, T. Impact of rule-violating behaviors on the risk of bus drivers being at-fault in crashes. Traffic Inj Prev 2022, 23, 364-368, doi:10.1080/15389588.2022.2079639.

5. Champahom, T.; Jomnonkwao, S.; Banyong, C.; Nambulee, W.; Karoonsoontawong, A.; Ratanavaraha, V. Analysis of Crash Frequency and Crash Severity in Thailand: Hierarchical Structure Models Approach. Sustainability 2021, 13, doi:10.3390/su131810086.

6. Gopalakrishnan, S. A public health perspective of road traffic accidents. J Family Med Prim Care 2012, 1, 144-150, doi:10.4103/2249-4863.104987.

7. Hartling, L.; Wiebe, N.; Russell, K.; Petruk, J.; Spinola, C.; Klassen, T.P. Graduated driver licensing for reducing motor vehicle crashes among young drivers. Cochrane Database Syst Rev 2004, 10.1002/14651858.CD003300.pub2, Cd003300, doi:10.1002/14651858.CD003300.pub2.

8. Wang, S.Y.; Wu, K.F. Reducing intercity bus crashes through driver rescheduling. Accid Anal Prev 2019, 122, 25-35, doi:10.1016/j.aap.2018.09.019.

9. Stewart, B.T.; Yankson, I.K.; Afukaar, F.; Medina, M.C.; Cuong, P.V.; Mock, C. Road Traffic and Other Unintentional Injuries Among Travelers to Developing Countries. Med Clin North Am 2016, 100, 331-343, doi:10.1016/j.mcna.2015.07.011.

10. Nguyen, T.C.; Nguyen, M.H.; Armoogum, J.; Ha, T.T. Bus Crash Severity in Hanoi, Vietnam. Safety 2021, 7, doi:10.3390/safety7030065.

11. Abdollahzadeh Nasiri, A.S.; Rahmani, O.; Abdi Kordani, A.; Karballaeezadeh, N.; Mosavi, A. Evaluation of Safety in Horizontal Curves of Roads Using a Multi-Body Dynamic Simulation Process. Int J Environ Res Public Health 2020, 17, doi:10.3390/ijerph17165975.

12. Jones, A.P.; Haynes, R.; Harvey, I.M.; Jewell, T. Road traffic crashes and the protective effect of road curvature over small areas. Health & Place 2012, 18, 315-320, doi:https://doi.org/10.1016/j.healthplace.2011.10.008.

13. Polus, A.; Pollatschek, M.A.; Farah, H. Impact of infrastructure characteristics on road crashes on two-lane highways. Traffic Inj Prev 2005, 6, 240-247, doi:10.1080/15389580590969210.

---

## [Decision Letter · Decision Letter 1]

14 Oct 2022

PONE-D-22-22254R1Factors affecting bus accident severity in Thailand: A multinomial logit model

PLOS ONE

Dear Dr. Singkham,

Thank you for submitting your manuscript to PLOS ONE. After careful consideration, we feel that it has merit but does not fully meet PLOS ONE’s publication criteria as it currently stands. Therefore, we invite you to submit a revised version of the manuscript that addresses the points raised during the review process.

We look forward to receiving your revised manuscript.

Kind regards,

Yanyong Guo, Ph.D

Academic Editor

PLOS ONE

Journal Requirements:

Reviewers' comments:

Reviewer's Responses to Questions

**Comments to the Author**

1. If the authors have adequately addressed your comments raised in a previous round of review and you feel that this manuscript is now acceptable for publication, you may indicate that here to bypass the “Comments to the Author” section, enter your conflict of interest statement in the “Confidential to Editor” section, and submit your "Accept" recommendation.

Reviewer #1: All comments have been addressed

Reviewer #2: All comments have been addressed

2. Is the manuscript technically sound, and do the data support the conclusions?

Reviewer #1: Yes

Reviewer #2: Partly

3. Has the statistical analysis been performed appropriately and rigorously? 

Reviewer #1: Yes

Reviewer #2: Yes

4. Have the authors made all data underlying the findings in their manuscript fully available?

Reviewer #1: Yes

Reviewer #2: No

5. Is the manuscript presented in an intelligible fashion and written in standard English?

Reviewer #1: Yes

Reviewer #2: Yes

6. Review Comments to the Author

Reviewer #1: The authors have addressed all my comments. the method is sound and the findings were supported by the data.

Reviewer #2: First of all, thank you for your response to the questions I asked previously. I think the current version has been upgraded more fully and is closer to the level of publication, and in order to promote readability, I have the following small suggestions:

1) The quantities and percentages in Table 2 are repetitive expressions of the same information and it is recommended that one presentation be retained. Also, please be careful with the use of percentage signs; it is very uncomfortable not to have the whole table look like a percentage sign;

2) 0.1567 in the first row of Table 3 should be taken as 0.157;

3) Please deal with the percentage signs in Table 4, the whole table is now about percentages and the same expression looks very uncomfortable;

4) Please provide information on whether the data is publicly available;

5) Further changes to the format of the references are required.

7. PLOS authors have the option to publish the peer review history of their article (what does this mean?). If published, this will include your full peer review and any attached files.

Reviewer #1: No

Reviewer #2: No

---

## [Author Response · Author response to Decision Letter 1]

24 Oct 2022

Response to Reviewer 1 Comments

Comments and Suggestions for Authors

Reviewer #1: 

Point 1: 

The authors have addressed all my comments. the method is sound and the findings were supported by the data.

Response 1

Thank you for your very kind comments.

Response to Reviewer 2 Comments

Comments and Suggestions for Authors

Reviewer #2: 

First of all, thank you for your response to the questions I asked previously. I think the current version has been upgraded more fully and is closer to the level of publication, and in order to promote readability, I have the following small suggestions:

Point 1: 

1) The quantities and percentages in Table 2 are repetitive expressions of the same information and it is recommended that one presentation be retained. Also, please be careful with the use of percentage signs; it is very uncomfortable not to have the whole table look like a percentage sign;

Response 1

Thank you for your kind comments. We agreed with the reviewer. We deleted the quantities (number of accidents) and percentage signs in Table 2 as following;

Table 2 

Descriptive statistics (%) of risk factors from 2010 to 2019 in Thailand.

Variables Non-injury Injury Fatality Total

Bus driver factors 

 Excessive speed 

 Noa 32.1 40.6 27.3 31.5

 Yes 47.0 38.1 14.9 68.5

 Not obeying traffic rules 

 Noa 42.9 39.2 17.9 94.8

 Yes 30.9 32.9 36.2 5.2

 Abrupt driving 

 Noa 43.4 38.9 17.7 88.3

 Yes 33.5 39.1 27.4 11.7

 Drunk driving 

 Noa 42.3 40.0 18.7 99.7

 Yes 30.0 10.0 60.0 0.34

 Fallen asleep 

 Noa 43.6 38.0 18.4 94.4

 Yes 20.9 53.4 25.7 5.6

 Driving direction 

 Unknown a 55.9 27.5 16.6 13.9

 Right-way 41.7 38.8 19.5 59.1

 Wrong-way 36.5 44.9 18.6 27.0

Crash characteristics 

 Month 

 Januarya 40.5 40.9 18.6 11.3

 February 43.8 36.4 19.8 7.4

 March 40.4 36.5 23.1 8.9

 April 40.9 40.5 18.6 13.8

 May 48.1 35.7 16.2 7.4

 June 47.8 38.5 13.7 6.3

 July 41.1 40.1 18.8 6.9

 August 43.7 38.0 18.3 7.7

 September 52.7 35.5 11.8 5.8

 October 39.1 38.1 22.8 6.8

 November 43.6 40.4 16.0 7.5

 December 33.9 42.4 23.7 10.1

 Time of day 

 Morninga 40.2 41.0 18.8 50.4

 Afternoon/evening 47.2 37.7 15.1 26.1

 Night 41.5 35.6 22.9 23.5

 Region 

 Centrala 57.2 31.6 11.2 44.5

 Northern 25.4 44.0 30.6 8.0

 Northeast 30.3 46.5 23.3 22.5

 Eastern 40.1 39.2 20.7 7.8

 Western 26.1 46.6 27.3 5.5

 Southern 29.1 45.0 25.9 11.7

Environmental characteristics 

 Highways without a frontage road 

 Noa 60.2 30.9 8.9 25.4

 Yes 36.2 41.6 22.2 74.6

 Road horizontal alignment 

 Straighta 44.9 38.5 16.6 85.5

 Curved 27.4 41.0 31.6 13.4

 Sharply curved 18.2 45.4 36.4 1.1

 Road vertical alignment 

 Flata 44.1 38.3 17.6 92.6

 Upwardly curved 25.0 33.9 41.1 1.9

 Downwardly curved 27.8 33.3 38.9 1.2

 Sloped 15.2 56.0 28.8 4.3

 Intersection type 

 No Intersectiona 43.3 38.9 17.8 88.5

 Four-way 33.8 40.0 27.2 4.7

 T-shaped 34.9 38.8 26.3 4.4

 Y-shaped 26.9 38.5 34.6 0.9

 Others 37.8 37.8 24.4 1.5

 Median opening 

 Noa 43.1 38.5 18.4 92.6

 Yes 31.9 43.5 24.5 7.4

 Road junctions 

 Noa 42.7 38.6 18.7 98.1

 Yes 22.2 51.9 25.9 1.9

This table displays percentage. The total column contains percentages calculated across rows within one variable. The injury severity columns contain percentages calculated across columns within one row.

a Indicates referent category.

Page 7 Line 238

Point 2:

2. 0.1567 in the first row of Table 3 should be taken as 0.157;

Response 2

Thank you for your suggestions. We revised the value from 0.1567 in the first row of Table 3 to 0.157.

Page 10 Line 1213

Point 3:

3) Please deal with the percentage signs in Table 4, the whole table is now about percentages and the same expression looks very uncomfortable;

Response 3: 

We agreed with the reviewer and deleted the percentage signs in Table 4 as following;

Table 4 

The results of the marginal effects for bus accidents in Thailand (%)

Variables Non-injury Injury Fatality

Bus driver factors

Not obeying traffic rules -11.4 -8.4 19.7

Abrupt driving -11.5 -1.5 13.1

Drunk driving -11.5 -30.9 42.4

Fallen asleep -22.5 10.9 11.1

Right-way driving direction -10.0 10.7 -0.7

Wrong-way driving direction -17.2 18.2 -1.0

Crash characteristics

Afternoon/evening 6.3 -2.5 -0.4

Northern region -19.4 8.4 11.1

Northeast region -18.5 10.5 8.1

Eastern region -9.9 2.8 7.1

Western region -17.8 7.2 10.6

Southern region -18.0 8.8 9.2

Environmental characteristics

Highways without frontage roads -15.9 7.5 8.3

Curve roads -6.6 -2.1 8.7

Sloped roads -22.6 17.2 5.5

Median openings -7.9 4.1 3.9

Road junctions -16.0 14.3 1.7

Page 11 Line 307

Point 4:

4) Please provide information on whether the data is publicly available;

Response 4: 

Thank you for your recommendation. We provided the information on whether the data is publicly available as following;

“Data Availability

All relevant data are within the paper and its Supporting information files.”

Page 11 Line 1231

Point 5:

5) Further changes to the format of the references are required.

Response 5: 

Thank you for your recommendation. We revised the format of the references using PLOS style in the main text as following;

1. Kaplan S, Prato CG. Risk factors associated with bus accident severity in the United States: a generalized ordered logit model. J Safety Res. 2012;43(3):171-80. Epub 2012/09/15. doi: 10.1016/j.jsr.2012.05.003. PubMed PMID: 22974682.

2. Morency P, Strauss J, Pépin F, Tessier F, Grondines J. Traveling by Bus Instead of Car on Urban Major Roads: Safety Benefits for Vehicle Occupants, Pedestrians, and Cyclists. J Urban Health. 2018;95(2):196-207. doi: 10.1007/s11524-017-0222-6. PubMed PMID: 29500736.

3. Pearce T, Maunder DAC, Mbara TC, Babu DM, Rwebangira T. Bus Accidents in India, Nepal, Tanzania, and Zimbabwe. Transportation Research Record. 2000;1726(1):16-23. doi: 10.3141/1726-03.

4. Department of Land Transport. Bus accident data https://web.dlt.go.th/statistics/index.php2021 [cited October 14, 2021]. Available from: https://web.dlt.go.th/statistics/index.php.

5. WHO. Global Status Report on Road Safety 2018 https://www.who.int/thailand/news/detail/19-12-2018-launch-of-the-global-status-report-on-road-safety-2018-in-thailand2018 [cited October 14, 2021]. Available from: https://www.who.int/thailand/news/detail/19-12-2018-launch-of-the-global-status-report-on-road-safety-2018-in-thailand.

6. Feng S, Li Z, Ci Y, Zhang G. Risk factors affecting fatal bus accident severity: Their impact on different types of bus drivers. Accid Anal Prev. 2016;86:29-39. Epub 2015/10/30. doi: 10.1016/j.aap.2015.09.025. PubMed PMID: 26513334.

7. Barua U, Tay R. Severity of urban transit bus crashes in Bangladesh. Journal of Advanced Transportation. 2010;44(1):34-41. doi: https://doi.org/10.1002/atr.104.

8. Anund A, Dukic T, Thornthwaite S, Falkmer T. Is European school transport safe?—The need for a “door-to-door” perspective. European Transport Research Review. 2011;3(2):75-83. doi: 10.1007/s12544-011-0052-7.

9. Tetali S, Edwards P, Murthy GVS, Roberts I. Road traffic injuries to children during the school commute in Hyderabad, India: cross-sectional survey. Injury Prevention. 2016;22(3):171. doi: 10.1136/injuryprev-2015-041854.

10. Chimba D, Sando T, Kwigizile V. Effect of bus size and operation to crash occurrences. Accident Analysis & Prevention. 2010;42(6):2063-7. doi: https://doi.org/10.1016/j.aap.2010.06.018.

11. Razmpa E, Sadegh Niat K, Saedi B. Urban bus drivers' sleep problems and crash accidents. Indian J Otolaryngol Head Neck Surg. 2011;63(3):269-73. Epub 2012/07/04. doi: 10.1007/s12070-011-0235-5. PubMed PMID: 22754808; PubMed Central PMCID: PMCPMC3138945.

12. Blower D, Green P. Type of Motor Carrier and Driver History in Fatal Bus Crashes. Transportation Research Record: Journal of the Transportation Research Board. 2010;2194. doi: 10.3141/2194-05.

13. af Wåhlberg AE. Characteristics of low speed accidents with buses in public transport: part II. Accid Anal Prev. 2004;36(1):63-71. Epub 2003/10/24. doi: 10.1016/s0001-4575(02)00128-8. PubMed PMID: 14572828.

14. af Wåhlberg AE. Speed choice versus celeration behavior as traffic accident predictor. J Safety Res. 2006;37(1):43-51. Epub 2006/02/28. doi: 10.1016/j.jsr.2005.10.017. PubMed PMID: 16499929.

15. Af Wåhlberg AE. The relation of non-culpable traffic incidents to bus drivers' celeration behavior. J Safety Res. 2008;39(1):41-6. Epub 2008/03/08. doi: 10.1016/j.jsr.2007.10.009. PubMed PMID: 18325415.

16. Damsere-Derry J, Adanu EK, Ojo TK, Sam EF. Injury-severity analysis of intercity bus crashes in Ghana: A random parameters multinomial logit with heterogeneity in means and variances approach. Accid Anal Prev. 2021;160:106323. Epub 2021/08/12. doi: 10.1016/j.aap.2021.106323. PubMed PMID: 34380083.

17. Jayatilleke AU, Nakahara S, Dharmaratne SD, Jayatilleke AC, Poudel KC, Jimba M. Working conditions of bus drivers in the private sector and bus crashes in Kandy district, Sri Lanka: a case-control study. Inj Prev. 2009;15(2):80-6. Epub 2009/04/07. doi: 10.1136/ip.2008.018937. PubMed PMID: 19346419.

18. Chaiard J, Deeluea J, Suksatit B, Songkham W. Factors associated with sleep quality of Thai intercity bus drivers. Industrial Health. 2019;57(5):596-603. doi: 10.2486/indhealth.2018-0168.

19. Prato CG, Kaplan S. Bus accident severity and passenger injury: evidence from Denmark. European Transport Research Review. 2014;6(1):17-30. doi: 10.1007/s12544-013-0107-z.

20. Ponnaluri RV. The odds of wrong-way crashes and resulting fatalities: A comprehensive analysis. Accident Analysis & Prevention. 2016;88:105-16. doi: https://doi.org/10.1016/j.aap.2015.12.012.

21. Riyapan S, Thitichai P, Chaisirin W, Nakornchai T, Chakorn T. Outcomes of Emergency Medical Service Usage in Severe Road Traffic Injury during Thai Holidays. West J Emerg Med. 2018;19(2):266-75. Epub 2018/02/20. doi: 10.5811/westjem.2017.11.35169. PubMed PMID: 29560053.

22. Klinjun N, Lim A, Bundhamcharoen K. Epidemiologic patterns of transport accident mortality in Thailand. Southeast Asian J Trop Med Public Health. 2016;47(2):318-27. Epub 2016/06/02. PubMed PMID: 27244970.

23. Abrari Vajari M, Aghabayk K, Sadeghian M, Shiwakoti N. A multinomial logit model of motorcycle crash severity at Australian intersections. Journal of Safety Research. 2020;73:17-24. doi: https://doi.org/10.1016/j.jsr.2020.02.008.

24. Khorshidi A, Ainy E, Hashemi Nazari SS, Soori H. Temporal Patterns of Road Traffic Injuries in Iran. Arch Trauma Res. 2016;5(2):e27894. Epub 2016/10/06. doi: 10.5812/atr.27894. PubMed PMID: 27703958; PubMed Central PMCID: PMCPMC5037289.

25. Shahla F, Shalaby AS, Persaud BN, Hadayeghi A. Analysis of Transit Safety at Signalized Intersections in Toronto, Ontario, Canada. Transportation Research Record. 2009;2102(1):108-14. doi: 10.3141/2102-14.

26. Nguyen TC, Nguyen MH, Armoogum J, Ha TT. Bus Crash Severity in Hanoi, Vietnam. Safety. 2021;7(3). doi: 10.3390/safety7030065.

27. Ahmed M, Khanom K, Shampa RM, Bari MH. Road traffic accident among motor vehicle drivers in selected high ways. Mymensingh Med J. 2004;13(2):165-8. Epub 2004/07/31. PubMed PMID: 15284694.

28. Yang J, Peek-Asa C, Cheng G, Heiden E, Falb S, Ramirez M. Incidence and characteristics of school bus crashes and injuries. Accident Analysis & Prevention. 2009;41(2):336-41. doi: https://doi.org/10.1016/j.aap.2008.12.012.

29. Abellán J, López G, de Oña J. Analysis of traffic accident severity using Decision Rules via Decision Trees. Expert Systems with Applications. 2013;40(15):6047-54. doi: https://doi.org/10.1016/j.eswa.2013.05.027.

30. de Lapparent M. Empirical Bayesian analysis of accident severity for motorcyclists in large French urban areas. Accident Analysis & Prevention. 2006;38(2):260-8. doi: https://doi.org/10.1016/j.aap.2005.09.001.

31. Department of Highways. Bus accident data https://www.doh.go.th/visual/accident2021 [cited October 14, 2021]. Available from: https://www.doh.go.th/visual/accident.

32. Champahom T, Jomnonkwao S, Banyong C, Nambulee W, Karoonsoontawong A, Ratanavaraha V. Analysis of Crash Frequency and Crash Severity in Thailand: Hierarchical Structure Models Approach. Sustainability. 2021;13(18). doi: 10.3390/su131810086.

33. Zimmerman K, Mzige AA, Kibatala PL, Museru LM, Guerrero A. Road traffic injury incidence and crash characteristics in Dar es Salaam: a population based study. Accid Anal Prev. 2012;45:204-10. Epub 2012/01/25. doi: 10.1016/j.aap.2011.06.018. PubMed PMID: 22269502.

34. Musa MF, Hassan SA, Mashros N. The impact of roadway conditions towards accident severity on federal roads in Malaysia. PLoS One. 2020;15(7):e0235564-e. doi: 10.1371/journal.pone.0235564. PubMed PMID: 32628689.

35. Chen Y, Wang K, King M, He J, Ding J, Shi Q, et al. Differences in Factors Affecting Various Crash Types with High Numbers of Fatalities and Injuries in China. PLoS One. 2016;11(7):e0158559. doi: 10.1371/journal.pone.0158559.

36. Wang Z, Huang B, Lu J, Zhao J, Zhang Y. Exploring the Impact of Access Designs on Crash Injury Severity on Multilane Highways. Traffic Injury Prevention. 2014;15(1):102-9. doi: 10.1080/15389588.2013.789871.

37. Vorko-Jović A, Kern J, Biloglav Z. Risk factors in urban road traffic accidents. J Safety Res. 2006;37(1):93-8. Epub 2006/03/07. doi: 10.1016/j.jsr.2005.08.009. PubMed PMID: 16516927.

38. Islam S, Mannering F. Driver aging and its effect on male and female single-vehicle accident injuries: Some additional evidence. Journal of Safety Research. 2006;37(3):267-76. doi: https://doi.org/10.1016/j.jsr.2006.04.003.

39. Savolainen P, Mannering F. Probabilistic models of motorcyclists’ injury severities in single- and multi-vehicle crashes. Accident Analysis & Prevention. 2007;39(5):955-63. doi: https://doi.org/10.1016/j.aap.2006.12.016.

40. Geedipally SR, Turner PA, Patil S. Analysis of Motorcycle Crashes in Texas with Multinomial Logit Model. Transportation Research Record. 2011;2265(1):62-9. doi: 10.3141/2265-07.

41. Çelik AK, Oktay E. A multinomial logit analysis of risk factors influencing road traffic injury severities in the Erzurum and Kars Provinces of Turkey. Accident Analysis & Prevention. 2014;72:66-77. doi: https://doi.org/10.1016/j.aap.2014.06.010.

42. Salum JH, Kitali AE, Bwire H, Sando T, Alluri P. Severity of motorcycle crashes in Dar es Salaam, Tanzania. Traffic Injury Prevention. 2019;20(2):189-95. doi: 10.1080/15389588.2018.1544706.

43. Withers SD. Data Analysis, Categorical. In: Kobayashi A, editor. International Encyclopedia of Human Geography (Second Edition). Oxford: Elsevier; 2009. p. 159-65.

44. Hashimoto EM, Ortega EMM, Cordeiro GM, Suzuki AK, Kattan MW. The multinomial logistic regression model for predicting the discharge status after liver transplantation: estimation and diagnostics analysis. J Appl Stat. 2020;47(12):2159-77. Epub 2019/12/24. doi: 10.1080/02664763.2019.1706725. PubMed PMID: 35706842; PubMed Central PMCID: PMCPMC9041638.

45. Rifaat SM, Tay R, de Barros A. Effect of street pattern on the severity of crashes involving vulnerable road users. Accident Analysis & Prevention. 2011;43(1):276-83. doi: https://doi.org/10.1016/j.aap.2010.08.024.

46. Savolainen PT, Mannering FL, Lord D, Quddus MA. The statistical analysis of highway crash-injury severities: A review and assessment of methodological alternatives. Accident Analysis & Prevention. 2011;43(5):1666-76. doi: https://doi.org/10.1016/j.aap.2011.03.025.

47. Hausman J, McFadden D. Specification Tests for the Multinomial Logit Model. Econometrica. 1984;52(5):1219-40. doi: 10.2307/1910997.

48. Yagoub U, Saiyed NS, Rahim B-EEA, Musawa N, Al Zahrani AM. Road Traffic Injuries and Related Safety Measures: A Multicentre Analysis at Military Hospitals in Tabuk, Saudi Arabia. Emerg Med Int. 2021;2021:6617381-. doi: 10.1155/2021/6617381. PubMed PMID: 33708446.

49. Sam EF, Daniels S, Brijs K, Brijs T, Wets G. Modelling public bus/minibus transport accident severity in Ghana. Accid Anal Prev. 2018;119:114-21. Epub 2018/07/18. doi: 10.1016/j.aap.2018.07.008. PubMed PMID: 30016751.

50. CTN NEWS. Thailand’s Transport Ministry to Ban Bus Drivers from Alcohol Consumption 2017 [27 April 2022]. Available from: https://www.chiangraitimes.com/news-asia-thailand/thailands-transport-ministry-to-ban-bus-drivers-from-alcohol-consumption/.

51. Hussain Q, Feng H, Grzebieta R, Brijs T, Olivier J. The relationship between impact speed and the probability of pedestrian fatality during a vehicle-pedestrian crash: A systematic review and meta-analysis. Accid Anal Prev. 2019;129:241-9. Epub 2019/06/09. doi: 10.1016/j.aap.2019.05.033. PubMed PMID: 31176144.

52. Sabbagh-Ehrlich S, Friedman L, Richter ED. Working conditions and fatigue in professional truck drivers at Israeli ports. Inj Prev. 2005;11(2):110-4. Epub 2005/04/05. doi: 10.1136/ip.2004.007682. PubMed PMID: 15805441; PubMed Central PMCID: PMCPMC1730197.

53. Anund A, Ihlström J, Fors C, Kecklund G, Filtness A. Factors associated with self-reported driver sleepiness and incidents in city bus drivers. Industrial health. 2016;54(4):337-46. Epub 2016/04/19. doi: 10.2486/indhealth.2015-0217. PubMed PMID: 27098307.

54. Rey de Castro J, Gallo J, Loureiro H. [Tiredness and sleepiness in bus drivers and road accidents in Peru: a quantitative study]. Rev Panam Salud Publica. 2004;16(1):11-8. Epub 2004/08/31. doi: 10.1590/s1020-49892004000700002. PubMed PMID: 15333261.

55. Kemel E. Wrong-way driving crashes on French divided roads. Accid Anal Prev. 2015;75:69-76. Epub 2014/12/03. doi: 10.1016/j.aap.2014.11.002. PubMed PMID: 25460093.

56. Gopalakrishnan S. A public health perspective of road traffic accidents. J Family Med Prim Care. 2012;1(2):144-50. Epub 2012/07/01. doi: 10.4103/2249-4863.104987. PubMed PMID: 24479025; PubMed Central PMCID: PMCPMC3893966.

57. Hartling L, Wiebe N, Russell K, Petruk J, Spinola C, Klassen TP. Graduated driver licensing for reducing motor vehicle crashes among young drivers. Cochrane Database Syst Rev. 2004;(2):Cd003300. Epub 2004/04/24. doi: 10.1002/14651858.CD003300.pub2. PubMed PMID: 15106200.

58. Abdul Manan MM, Várhelyi A, Çelik AK, Hashim HH. Road characteristics and environment factors associated with motorcycle fatal crashes in Malaysia. IATSS Research. 2018;42(4):207-20. doi: https://doi.org/10.1016/j.iatssr.2017.11.001.

59. Dechkamfoo C, Sitthikankun S, Kridakorn Na Ayutthaya T, Manokeaw S, Timprae W, Tepweerakun S, et al. Impact of Rainfall-Induced Landslide Susceptibility Risk on Mountain Roadside in Northern Thailand. Infrastructures. 2022;7(2). doi: 10.3390/infrastructures7020017.

60. Wang SY, Wu KF. Reducing intercity bus crashes through driver rescheduling. Accid Anal Prev. 2019;122:25-35. Epub 2018/10/10. doi: 10.1016/j.aap.2018.09.019. PubMed PMID: 30300796.

61. Stewart BT, Yankson IK, Afukaar F, Medina MC, Cuong PV, Mock C. Road Traffic and Other Unintentional Injuries Among Travelers to Developing Countries. Med Clin North Am. 2016;100(2):331-43. Epub 2016/02/24. doi: 10.1016/j.mcna.2015.07.011. PubMed PMID: 26900117; PubMed Central PMCID: PMCPMC4764791.

62. Milton JC, Shankar VN, Mannering FL. Highway accident severities and the mixed logit model: an exploratory empirical analysis. Accid Anal Prev. 2008;40(1):260-6. Epub 2008/01/25. doi: 10.1016/j.aap.2007.06.006. PubMed PMID: 18215557.

63. Buddhavarapu P, Banerjee A, Prozzi JA. Influence of pavement condition on horizontal curve safety. Accid Anal Prev. 2013;52:9-18. Epub 2013/01/10. doi: 10.1016/j.aap.2012.12.010. PubMed PMID: 23298704.

64. Chang F, Li M, Xu P, Zhou H, Haque MM, Huang H. Injury Severity of Motorcycle Riders Involved in Traffic Crashes in Hunan, China: A Mixed Ordered Logit Approach. Int J Environ Res Public Health. 2016;13(7):714. doi: 10.3390/ijerph13070714. PubMed PMID: 27428987.

65. Zhao XG, He XD, Wu JS, Zhao GF, Ma YF, Zhang M, et al. Risk factors for urban road traffic injuries in Hangzhou, China. Arch Orthop Trauma Surg. 2009;129(4):507-13. Epub 2009/02/18. doi: 10.1007/s00402-009-0827-7. PubMed PMID: 19221774.

66. Abdollahzadeh Nasiri AS, Rahmani O, Abdi Kordani A, Karballaeezadeh N, Mosavi A. Evaluation of Safety in Horizontal Curves of Roads Using a Multi-Body Dynamic Simulation Process. Int J Environ Res Public Health. 2020;17(16). Epub 2020/08/23. doi: 10.3390/ijerph17165975. PubMed PMID: 32824601; PubMed Central PMCID: PMCPMC7459981.

67. Jones AP, Haynes R, Harvey IM, Jewell T. Road traffic crashes and the protective effect of road curvature over small areas. Health & Place. 2012;18(2):315-20. doi: https://doi.org/10.1016/j.healthplace.2011.10.008.

68. Polus A, Pollatschek MA, Farah H. Impact of infrastructure characteristics on road crashes on two-lane highways. Traffic Inj Prev. 2005;6(3):240-7. Epub 2005/08/10. doi: 10.1080/15389580590969210. PubMed PMID: 16087465.

69. Guo Y, Li Z, Wu Y, Xu C. Exploring unobserved heterogeneity in bicyclists’ red-light running behaviors at different crossing facilities. Accident Analysis & Prevention. 2018;115:118-27. doi: https://doi.org/10.1016/j.aap.2018.03.006.

70. Guo Y, Li Z, Wu Y, Xu C. Evaluating factors affecting electric bike users’ registration of license plate in China using Bayesian approach. Transportation Research Part F: Traffic Psychology and Behaviour. 2018;59:212-21. doi: https://doi.org/10.1016/j.trf.2018.09.008.

71. Alnawmasi N, Mannering F. A statistical assessment of temporal instability in the factors determining motorcyclist injury severities. Analytic Methods in Accident Research. 2019;22:100090. doi: https://doi.org/10.1016/j.amar.2019.100090.

72. Xie P, Qin D, Zhu T. Impact of rule-violating behaviors on the risk of bus drivers being at-fault in crashes. Traffic Inj Prev. 2022;23(6):364-8. Epub 2022/06/08. doi: 10.1080/15389588.2022.2079639. PubMed PMID: 35670548.

Page 16 Line 1545

---

## [Decision Letter · Decision Letter 2]

25 Oct 2022

Factors affecting bus accident severity in Thailand: A multinomial logit model

PONE-D-22-22254R2

Dear Dr. Singkham,

We’re pleased to inform you that your manuscript has been judged scientifically suitable for publication and will be formally accepted for publication once it meets all outstanding technical requirements.

Kind regards,

Yanyong Guo, Ph.D

Academic Editor

PLOS ONE

Additional Editor Comments (optional):

Reviewers' comments:

Reviewer's Responses to Questions

**Comments to the Author**

1. If the authors have adequately addressed your comments raised in a previous round of review and you feel that this manuscript is now acceptable for publication, you may indicate that here to bypass the “Comments to the Author” section, enter your conflict of interest statement in the “Confidential to Editor” section, and submit your "Accept" recommendation.

Reviewer #2: All comments have been addressed

2. Is the manuscript technically sound, and do the data support the conclusions?

Reviewer #2: Yes

3. Has the statistical analysis been performed appropriately and rigorously? 

Reviewer #2: Yes

4. Have the authors made all data underlying the findings in their manuscript fully available?

Reviewer #2: Yes

5. Is the manuscript presented in an intelligible fashion and written in standard English?

Reviewer #2: Yes

6. Review Comments to the Author

Reviewer #2: The author has made serious changes to address the comments and suggestions I made and now I have no further questions.

7. PLOS authors have the option to publish the peer review history of their article (what does this mean?). If published, this will include your full peer review and any attached files.

Reviewer #2: No
